# Bandgap Engineering of Two-Dimensional Double Perovskite Cs_4_AgBiBr_8_/WSe_2_ Heterostructure from Indirect Bandgap to Direct Bandgap by Introducing Se Vacancy

**DOI:** 10.3390/ma16103668

**Published:** 2023-05-11

**Authors:** Yiwei Cai, Zhengli Lu, Xin Xu, Yujia Gao, Tingting Shi, Xin Wang, Lingling Shui

**Affiliations:** 1School of Information and Optoelectronic Science and Engineering, South China Normal University, Guangzhou 510006, Chinashuill@m.scnu.edu.cn (L.S.); 2Siyuan Laboratory, Department of Physics, Jinan University, Guangzhou 510632, China; 3Guangzhou Key Laboratory of Vacuum Coating Technologies and New Energy Materials, Jinan University, Guangzhou 510632, China; 4International Academy of Optoelectronics at Zhaoqing, South China Normal University, Guangzhou 510006, China; 5Guangdong Provincial Key Laboratory of Nanophotonic Functional Materials and Device, South China Normal University, Guangzhou 510006, China

**Keywords:** van der Waals heterostructure, first-principles, electronic properties, bandgap engineering

## Abstract

Heterostructures based on layered materials are considered next-generation photocatalysts due to their unique mechanical, physical, and chemical properties. In this work, we conducted a systematic first-principles study on the structure, stability, and electronic properties of a 2D monolayer WSe_2_/Cs_4_AgBiBr_8_ heterostructure. We found that the heterostructure is not only a type-II heterostructure with a high optical absorption coefficient, but also shows better optoelectronic properties, changing from an indirect bandgap semiconductor (about 1.70 eV) to a direct bandgap semiconductor (about 1.23 eV) by introducing an appropriate Se vacancy. Moreover, we investigated the stability of the heterostructure with Se atomic vacancy in different positions and found that the heterostructure was more stable when the Se vacancy is near the vertical direction of the upper Br atoms from the 2D double perovskite layer. The insightful understanding of WSe_2_/Cs_4_AgBiBr_8_ heterostructure and the defect engineering will offer useful strategies to design superior layered photodetectors.

## 1. Introduction

Organic–inorganic metal halide perovskites demonstrate high performance, but still face one major challenge: a stability issue due to hydrophilic organic cations and displaying very low thermal decomposition temperatures. Recently, all-inorganic halide perovskites, especially the double perovskite, without molecules and lead-free perovskite showing excellent stability against moisture, heat, and light, have attracted extensive attention [1,2,3]. In general, since the 2D all-inorganic double perovskite Cs_4_AgBiBr_8_ has a high exciton-binding energy, which is 3 times larger than that of the 3D all-inorganic double perovskite Cs_2_AgBiBr_6_, the charge carrier mobility and absorption coefficients of Cs_4_AgBiBr_8_ (2D) in the visible spectrum are worse than those of Cs_2_AgBiBr_6_ (3D) [4]. However, in 2021, Wang et al. reported the upconversion photovoltaic effect of WS_2_ monolayer/(C_6_H_5_C_2_H_4_NH_3_)_2_PbI_4_ 2D perovskite heterostructures by below-bandgap two-photon absorption via a virtual intermediate state, which enables heterojunction devices with good photoresponsivity and excellent current on/off ratio [5]. Therefore, heterostructure is considered an efficient way to enhance device performance, especially the charge mobility and light adsorption.

Furthermore, van der Waals heterostructures (vdWHs) based on 2DLMs (2D layered materials) with selectable material properties pave the way to build new structures at the atomic scale, which may lead to new heterostructures with novel physical properties and versatility [6,7]. Two-dimensional layered materials, such as graphene and transition metal dichalcogenides (TMDs), have attracted great attention due to their extraordinary properties in fundamental physics and potential applications [8,9]. Interestingly, MoX_2_ and WX_2_ (X = S, Se, Te) TMDs are indirect gap semiconductors in their bulk states, but some of them can become direct gap semiconductors when the film thickness is intentionally changed into a monolayer [10,11,12]. Due to the weakened dielectric shielding and layer-dependent electronic structure, monolayer TMDs such as MoS_2_ and WSe_2_ have direct band gaps with strong exciton characteristics in the visible or near-infrared range [13]. However, their relatively weak light absorption hinders their practical application. Fang et al. reported that integrating monolayer TMDs with 2D perovskites, which serve as the light-absorption layer, can be an efficient solution. Through the complementary effect of two-dimensional perovskites, heterostructure engineering based on TMD layers can effectively improve the performance of photodetectors with low-power optoelectronic applications [14].

A TMD, WSe_2_, has recently received more attention [15,16,17]. Two-dimensional WSe_2_ thin films and Nanoflakes have been used for photoelectrochemical hydrogen production [18,19]. In this work, we used first-principles calculations to investigate the effect of WSe_2_ layer defects on the heterostructure properties of a monolayer WSe_2_/monolayer Cs_4_AgBiBr_8_ heterostructure. Our computational results show that the heterostructure exhibits an indirect band gap where the CBM (conduction band minimum) and VBM (valence band maximum) positions locate at different k points when the WSe_2_ monolayer is combined with the Cs_4_AgBiBr_8_ monolayer. Xia et al. reported that when the atom vacancy defect of vdWHs generates a flat defect energy level [20], we can employ the strategy of defect modification and successfully change the heterostructure from an indirect band gap to a direct band gap by introducing the defect energy level. Finally, the obtained heterostructure with a Se vacancy exhibits a direct band gap, and the heterostructure is more stable when the Se vacancy is near the vertical direction of the upper Br atoms from the 2D double perovskite layer.

## 2. Calculation Methods

Our first-principles calculations were performed using density functional theory (DFT) [21,22], which is implemented in the VASP code [23,24] with the standard frozen-core projector-augmented wave (PAW) [25] method. We used the generalized gradient approximation (GGA) [26] by the Perdew–Burke–Ernzerhof (PBE) method [27] to relax the structure. The cut-off energy is set to 400 eV. We used a 2 × 3 × 1 k-points Γ-centered mesh for calculating total energy and the structure relaxations of the WSe_2_/Cs_4_AgBiBr_8_ heterostructure, WSe_2_/Cs_4_AgBiBr_8_ heterostructure with W vacancy, and WSe_2_/Cs_4_AgBiBr_8_ heterostructure with Se vacancy. A Γ-centered mesh of 9 × 9 × 2 k-points was used for the calculations of the WSe_2_ monolayer. All structures were relaxed until the energy was less than 10^−5^ eV per atom and the force on each ion reduced below 0.03 eV Å^−1^. Then the electronic properties were calculated with the optimized structures. Considering the underestimation of the GGA–PBE functional of the band gaps, we employed the Heyd–Scuseria–Ernzerhof (HSE06) hybrid functional [28] to calculate the electronic structure of the WSe_2_ monolayer and Cs_4_AgBiBr_6_ monolayer for accurate band gap value.

## 3. Results and Discussion

### 3.1. Construction and Stability of the Heterostructure

We first investigated the structure of the WSe_2_ monolayer and Cs_4_AgBiBr_8_. The WSe_2_ monolayer is built by the monolayer 2D structure separated from WSe_2_ bulk (space group: *P*6_3_*mmc* and lattice parameters: *a* = 3.327 Å, *c* = 15.069 Å from [29,30]) and adding a vacuum layer of 15 Å. We analyzed the WSe_2_ monolayer using the GGA–PBE method; the space group is *P*$\overset{-}{6}$*m*2, and the optimized lattice parameters are *a* = 3.315 Å and *c* = 18.031 Å. We constructed a single-layer two-dimensional double perovskite Cs_4_AgBiBr_8_ by cutting the (0 1 1) surface of the double perovskite Cs_2_AgBiBr_6_ (space group: *Fm*$\overset{-}{3}$*m*; lattice parameters: *a* = 8.123 Å). To ensure that the analysis object is the Cs_4_AgBiBr_8_ monolayer, we added a vacuum layer of 15 Å [31] to the unit cell of Cs_4_AgBiBr_8_ to avoid the effect of periodicity. We analyzed the WSe_2_ monolayer with the GGA–PBE method; the space group is *P*4*mmm*, and the optimized lattice parameters are *a* = 8.123 Å *c* = 20.782 Å.

Heterostructures were built by the Structural Utilities (804) of ‘vaspkit’ [32] to find two representative heterostructures containing 73 atoms (containing 8 Cs, 2 Ag, 2 Bi, 16 Br, 15 W, and 30 Se atoms) with a suitable lattice shape and reasonable lattice mismatch respectively, named Heterostructure A and Heterostructure B (shown in Figure 1). The lattice mismatch rate σ (the lattice mismatch which can quantify structural match of crystals) is 2.027% and 4.048%, respectively, less than 5%. Heterostructure A is obtained by expanding the WSe_2_ single crystal according to the transformation matrix $\left[\begin{array}{ccc} 0 & 5 & 0\\ 3 & 4 & 0\\ 0 & 0 & 1 \end{array}\right]$, and the supercell is obtained by the Cs_4_AgBiBr_8_ single crystal expanding according to the transformation matrix $\left[\begin{array}{ccc} 0 & 2 & 0\\ 1 & 1 & 0\\ 0 & 0 & 1 \end{array}\right]$. Heterostructure B is obtained by expanding the WSe_2_ single crystal according to the transformation matrix $\left[\begin{array}{ccc} 1 & 4 & 0\\ 5 & 5 & 0\\ 0 & 0 & 1 \end{array}\right]$, and the supercell is obtained by the Cs_4_AgBiBr_8_ single crystal expanding according to the transformation matrix $\left[\begin{array}{ccc} 1 & 1 & 0\\ 0 & 2 & 0\\ 0 & 0 & 1 \end{array}\right]$. Following several attempts, the vacuum layer of the two species in the heterostructure is set to 3.5 Å. In order to avoid the influence of periodicity on the calculation results, a vacuum layer of 15 Å is set outside the heterostructure.

The two types of heterostructure are very similar; Heterostructure B can be obtained from Heterostructure A when the W atom and the vertical two Se atoms are exchanged in the horizontal direction. Interestingly, in the vertical direction, the atoms position correspondences are different; the calculated results show that the properties of the two heterostructures are basically the same. In order to quantify the thermodynamic stability of the interaction between WSe_2_ and Cs_4_AgBiBr_8_, the interface adhesion energy $E_{ad}$, a good descriptor, is obtained according to Equation (1):(1)$$E_{ad}=E_{hete.}-E_{W{Se}_{2}}-E_{{Cs}_{4}AgBi{Br}_{8},}$$
where $E_{hete.}$, $E_{W{Se}_{2}},$ and $E_{{Cs}_{4}AgBi{Br}_{8}}$ represent the total energies of the relaxed heterostructures WSe_2_/Cs_4_AgBiBr_8_, monolayer WSe_2_, and monolayer Cs_4_AgBiBr_8_, respectively. We calculated the adhesion energy of the two heterostructures to be −0.226 eV and −0.219 eV, respectively. Obviously, Heterostructure A has lower adhesion energy and lattice mismatch rate, so we decided to use Heterostructure A for follow-up research.

After geometric optimization, the surface of WSe_2_ did not exhibit significant deformation. The WSe_2_/Cs_4_AgBiBr_8_ heterostructure has a typical vdW equilibrium spacing, which was calculated to be 3.67 Å (Figure 1a) between the WSe_2_ and Cs_4_AgBiBr_8_ layers. We noticed that when WSe_2_ was adsorbed onto the surface of Cs_4_AgBiBr_8_(001), only physical adsorption occurred, but no chemical adsorption was observed.

### 3.2. Electronic Properties of WSe_2_/Cs_4_AgBiBr_8_

To investigate the photocatalytic performance of the WSe_2_/Cs_4_AgBiBr_8_ heterostructure, the energy band structures of the WSe_2_ monolayer, Cs_4_AgBiBr_8_ monolayer, and WSe_2_/Cs_4_AgBiBr_8_ heterostructure were calculated, respectively. We employed GGA–PBE and HSE06 methods to acquire more exact electronic structure properties, as shown in Figure 2. The WSe_2_ monolayer has a direct bandgap of 1.55 eV with the GGA–PBE method, which agrees well with the 1.6 eV estimated by extrapolating, as shown in Figure 2a. For the WSe_2_ monolayer, CBM and VBM are located at the K point, while the Cs_4_AgBiBr_8_ monolayer has an indirect bandgap of 2.09 eV, as shown in Figure 2b. With the HSE06 method, the WSe_2_ monolayer has a direct bandgap of 2.03 eV, and the bandgap of Cs_4_AgBiBr_8_ monolayer is 3.23 eV. Thus, the GGA–PBE functional might underestimate the band gaps of the heterostructures that we built, by about 0.6 eV, at least.

The WSe_2_/Cs_4_AgBiBr_8_ heterostructure is a type-II heterostructure which has better photocatalytic performance, owing to its effectiveness in spatially separating photogenerated electron-hole pairs by band alignment between two semiconductors [33]. By comparing the band structures of the Cs_4_AgBiBr_8_ monolayer and the WSe_2_ monolayer, we found that the band structure of the WSe_2_/Cs_4_AgBiBr_8_ heterostructure is influenced by vdW interaction between Cs_4_AgBiBr_8_ and WSe_2_ interfaces, rather than an uncomplicated superposition of the WSe_2_ monolayer and the Cs_4_AgBiBr_8_ monolayer. The energy bands of WSe_2_/Cs_4_AgBiBr_8_ heterostructures are staggered around the forbidden band, as shown in Figure 3a, in which the red and blue parts are contributed by Cs_4_AgBiBr_8_ and WSe_2_, respectively. The band structure of the WSe_2_/Cs_4_AgBiBr_8_ heterostructure has an indirect bandgap of 1.10 eV using the GGA–PBE method, which did not meet our expectations. The VBM and CBM of the heterostructure were located at Γ point and the point between the Γ point and X point, respectively. Obviously, the Cs_4_AgBiBr_8_ monolayer contributes the VBM, and the CBM comes from the WSe_2_ monolayer. Learning from the total and atom-projected density of states (TDOS and PDOS) between −1.5 eV and 1.5 eV, as shown in Figure 3a, the VBM is mainly composed of Ag d state and Br p state; at the same time, the CBM is mainly composed of W d state and Se p state.

The VBM of WSe_2_/Cs_4_AgBiBr_8_ heterostructure is provided by the Cs_4_AgBiBr_8_ layer, while the CBM is provided by the WSe_2_ layer. This demonstrates that the holes and excited electrons are separately confined to different layers of the heterostructure, which promotes the formation of spatially indirect excitons.

### 3.3. Cs_4_AgBiBr_8_/WSe_2_ with Defects

#### 3.3.1. Cs_4_AgBiBr_8_/WSe_2_ with W Vacancy

It can be learned from Figure 3c that W atoms play a major role in the CBM, so the effect of adding W vacancies to the heterostructure is also considered. The WSe_2_ layer consists of 15 WSe_2_ molecules with a total of 15 W atoms and 30 Se atoms. We attempted to analyze 15 different heterostructures with W atom vacancies using the GGA–PBE method (one of which is shown in Appendix A). Electronic properties of 15 W atom vacancy heterostructures are similar and not ideal. It can be learned from the energy band structures and the DOS (Figure 3c) that the defect states brought by the W vacancies are generated near the VBM and coincide with the VBM, which makes both sides of the forbidden band provided by WSe_2_ and the band gap is still indirect, which is not conducive to improving the photocatalytic activity.

#### 3.3.2. Stability of Cs_4_AgBiBr_8_/WSe_2_ with Se Vacancy

WSe_2_/Cs_4_AgBiBr_8_ heterostructure is an indirect band gap heterostructure, which is not conducive to improving the photocatalytic activity, and the positions of VBM and CBM are very close, so we tried to introduce defects in the heterostructure. The VBM provided by Cs_4_AgBiBr_8_ is located at the Γ point, while the CBM provided by WSe_2_ is located between the Γ point and the X point, not on the ordinary high symmetry point. Compared with the 2D double perovskite Cs_4_AgBiBr_8_ layer, the WSe_2_ layer has better ductility. Combining the above two points, we decided to study the WSe_2_ layers in WSe_2_/Cs_4_AgBiBr_8_, namely the W vacancy and the Se vacancy defects.

Interestingly, we found that the introduction of Se atomic vacancy evidently affects the electronic properties of the heterostructures. We numbered the 30 Se atoms in the WSe_2_ layer as 1 to 30 (as shown in Figure 4a), where the odd numbers are the upper layer, and the even numbers are the lower layer. The calculated results show that all 30 Cs_4_AgBiBr_8_/WSe_2_ heterostructures with Se atom vacancies are very close in electronic properties and all have direct band gaps, two of which are shown in Appendix A. Compared to the original heterostructure, that with Se atom vacancies shows minimal structure change. We therefore calculated the adhesion energies of 30 heterostructures with Se atom vacancy (Figure 4b). We found that the adhesion energy of most of the 30 heterostructures with Se atom vacancy is around −0.225 eV, and the heterostructures have lower interfacial adhesion energy when the defects appear on the lower surface of the WSe_2_ layer. Clearly, we observe that the heterostructures with No. 6 Se vacancy and the No. 14 Se vacancy have abnormally low adhesion energies. Both of the heterostructures with No. 6 Se vacancy and No. 14 Se vacancy are near the vertical direction of the upper Br atoms from the 2D double perovskite layer. The abnormally low interfacial adhesion energy may be caused by this Br atom.

#### 3.3.3. Electronic Structure of Cs_4_AgBiBr_8_/WSe_2_ with Se Vacancy

We analyzed Cs_4_AgBiBr_8_/WSe_2_ heterostructures with No. 6 Se vacancy (shown in Appendix A), which has the lowest adhesion energy. We found a defect level provided by WSe_2_ between the Fermi level and the conduction band, which is quite flat and reaches a minimum at the Γ point (Figure 3b). Compared with the Cs_4_AgBiBr_8_/WSe_2_ heterostructure energy band structure (Figure 3a), the Se atom vacancy has little effect on the valence band, and the VBM is still located at the Γ point. Therefore, we speculate that the Cs_4_AgBiBr_8_/WSe_2_ heterostructure with Se atom vacancy has a direct band gap of 0.63 eV, and both VBM and CBM are located at the Γ point.

We calculated the partial charge densities for the CBM and VBM of the Cs_4_AgBiBr_8_/WSe_2_ heterostructure and of the heterostructure with Se vacancy (shown in Figure 4c–f, respectively). The electron orbitals of WSe_2_ and Cs_4_AgBiBr_8_ occupy the CBM and VBM of the Cs_4_AgBiBr_8_/WSe_2_ heterostructure, respectively, which is consistent with the earlier analysis about DOS. Comparing the partial charge densities of CBM between the original heterostructure and heterostructure with Se vacancy, it is evident that the charge centers on the Se vacancy, which is uniformly distributed in the original heterostructure. The partial charge density of VBM has no change when there is a Se vacancy in the Cs_4_AgBiBr_8_/WSe_2_ heterostructure.

To understand how charges are transferring at the interface, we calculated the work function, which is used as an intrinsic reference for band alignment, of the Cs_4_AgBiBr_8_ monolayer, WSe_2_ monolayer with Se vacancy, and Cs_4_AgBiBr_8_/WSe_2_ with Se vacancy using the GGA–PBE method. The work function is calculated according to Equation (2) [34]:(2)$$\Phi =E_{vac}-E_{fermi},$$
where Φ, $E_{vac}$, and $E_{fermi}$ represent the work function, the electrostatic potential of the vacuum level, and the Fermi level, respectively. Based on Equation (2), the work functions of the Cs_4_AgBiBr_8_ monolayer, WSe_2_ monolayer with Se vacancy, and Cs_4_AgBiBr_8_/WSe_2_ with Se vacancy are 4.33 eV, 5.07 eV, and 4.43 eV, respectively, as shown in Figure 5a–c.

It can be seen from the electrostatic potentials that the electrons in the Cs_4_AgBiBr_8_ layer with low work function flow into the WSe_2_ layer, which has high work function after the electrostatic potential contact is formed. Therefore, the negative charges will accumulate at the interface of the WSe_2_ layer; at the same time, the positive charges will accumulate at the interface of the Cs_4_AgBiBr_8_ layer. Finally, the two Fermi levels of WSe_2_ and Cs_4_AgBiBr_8_ reach the same energy level; then, an internal electric field, which is generated by this spontaneous interfacial charge transfer, takes shape at the interface from the WSe_2_ layer to the Cs_4_AgBiBr_8_ layer. We plotted the energy level lineup diagrams and the charge separation schematic of the Cs_4_AgBiBr_8_ monolayer and WSe_2_ monolayer before and after contact, as shown in Figure 5d.

## 4. Conclusions

We constructed van der Waals heterostructures using monolayer WSe_2_ and monolayer 2D Cs_4_AgBiBr_8_. Considering a reasonable lattice mismatch rate and interface adhesion energy, we constructed a newWSe_2_/Cs_4_AgBiBr_8_ heterostructure with a lattice mismatch rate of 2.027% and the lowest interface adhesion energy. Importantly, layered WSe_2_/Cs_4_AgBiBr_8_ can form a type-II heterostructure. By introducing Se vacancy, we further successfully converted it into a direct band gap WSe_2_/Cs_4_AgBiBr_8_ heterostructure. We also found the most stable WSe_2_/Cs_4_AgBiBr_8_ heterostructure with Se vacancy, when Se vacancy appeared on the lower surface of WSe_2_ and near the vertical direction of the upper Br atoms from the 2D double perovskite layer. These results indicate that it is possible to construct type-II van-der-Waals heterostructures composed of TMD monolayers and 2D double perovskites from indirect band gaps to direct band gaps based on bandgap engineering.

## Figures and Tables

**Figure 1 materials-16-03668-f001:**
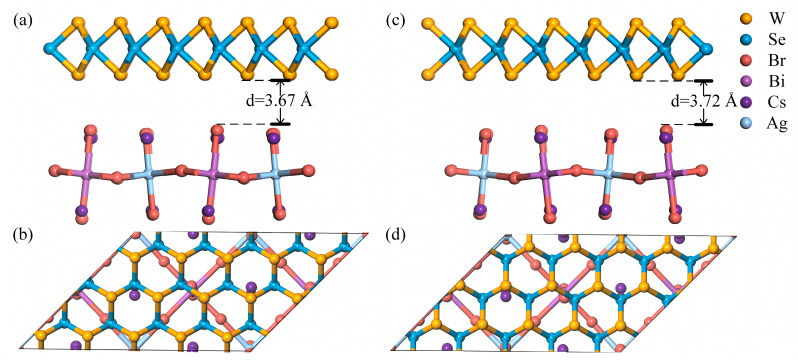
(**a**) Side and (**b**) top views of Heterostructure A; (**c**) side and (**d**) top views of Heterostructure B.

**Figure 2 materials-16-03668-f002:**
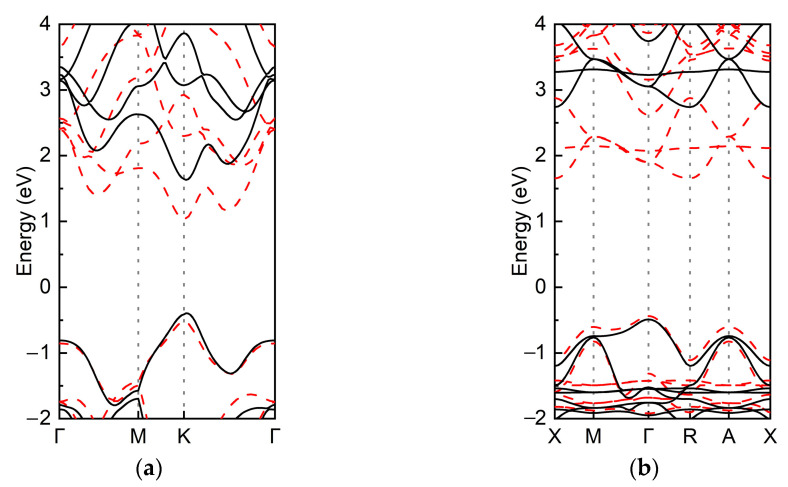
Energy band structures of the (**a**) WSe_2_ monolayer and (**b**) Cs_4_AgBiBr_8_ monolayer with PBE (red dashed lines) and HSE (black solid lines) methods.

**Figure 3 materials-16-03668-f003:**
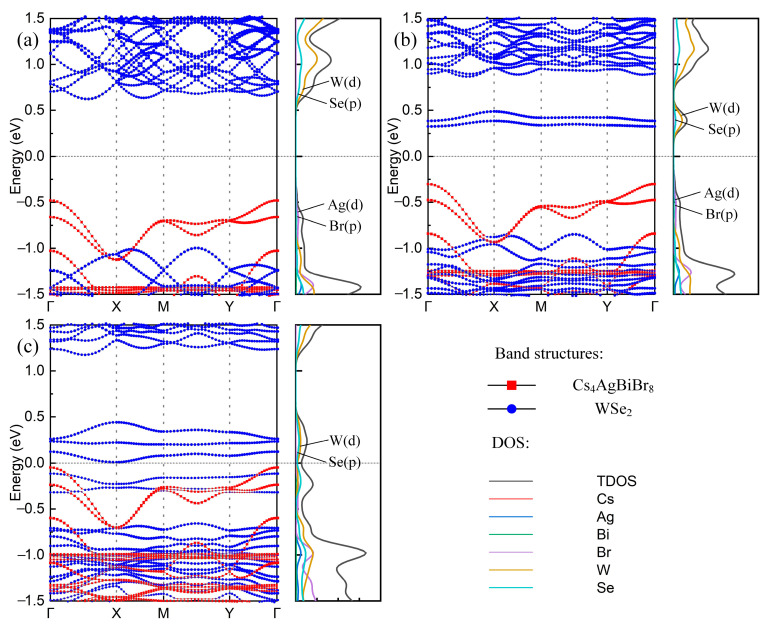
The projected band structures (**left**) and the DOS (**right**) of the (**a**) Cs_4_AgBiBr_8_/WSe_2_ heterostructure, (**b**) Cs_4_AgBiBr_8_/WSe_2_ heterostructure with No. 6 Se vacancy, and (**c**) Cs_4_AgBiBr_8_/WSe_2_ heterostructure with W vacancy. The red square and blue circle in the band structures show the electron orbits of Cs_4_AgBiBr_8_ and WSe_2_, respectively.

**Figure 4 materials-16-03668-f004:**
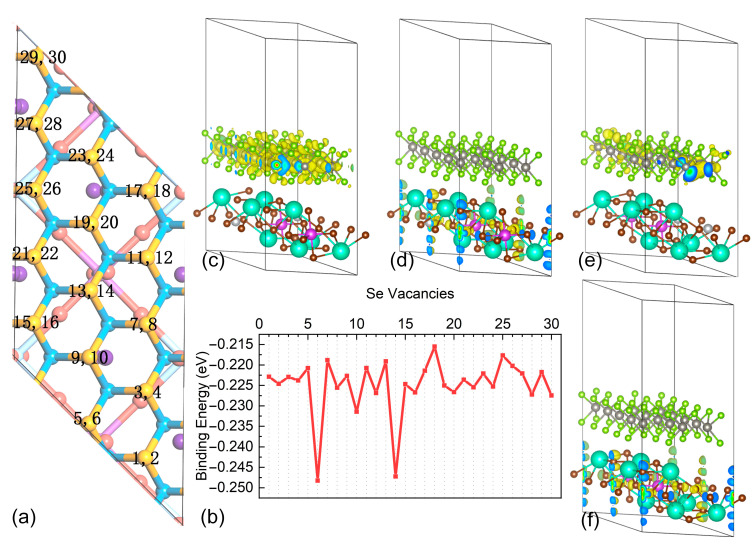
(**a**) Se atoms (n = 30) in the WSe_2_ layer of the heterostructure with Se atom vacancy defects. The atoms are numbered 1 to 30, where the odd numbers are the upper layer, and the even numbers are the lower layer. (**b**) The adhesion energy of Cs4AgBiBr_8_/WSe_2_ heterostructures with 30 Se atom vacancy defects (data are shown in Appendix A). The partial charge densities of the (**c**) CBM and (**d**) VBM of the Cs_4_AgBiBr_8_/WSe_2_ heterostructure, and that of the (**e**) CBM and (**f**) VBM of the Cs_4_AgBiBr_8_/WSe_2_ heterostructure with Se vacancy.

**Figure 5 materials-16-03668-f005:**
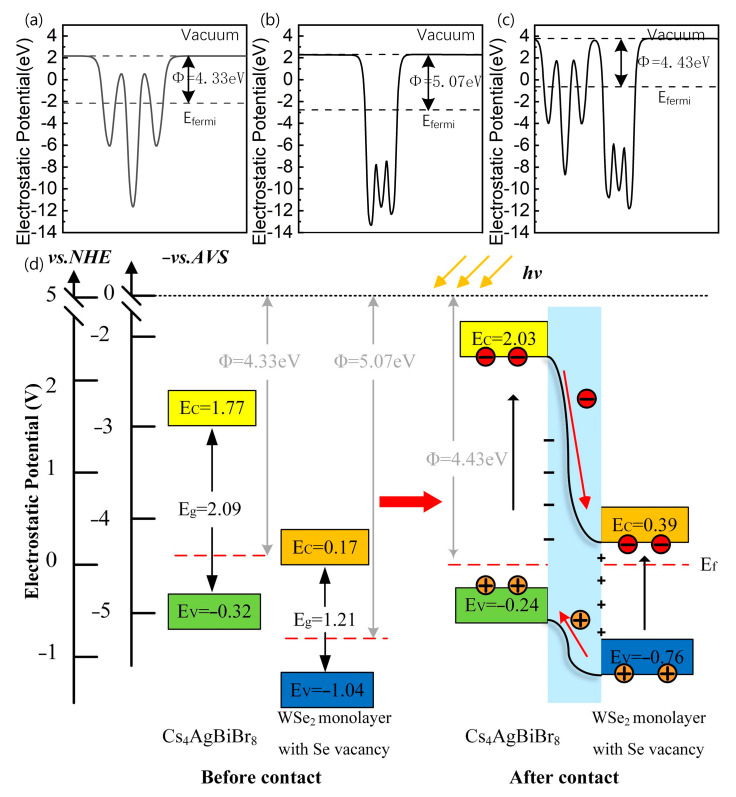
The electrostatic potentials of the (**a**) Cs_4_AgBiBr_8_ monolayer, (**b**) WSe_2_ monolayer with Se vacancy, and (**c**) Cs_4_AgBiBr_8_/WSe_2_ heterostructure with Se vacancy, respectively. (**d**) The band diagram of the WSe_2_/Cs_4_AgBiBr_8_ heterostructure and schematic of the charge separation at its interface under sunlight irradiation.

## Data Availability

The data that support the findings of this study are available from the corresponding author upon reasonable request.

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
