# Peer review of "Bandgap Engineering of Two-Dimensional Double Perovskite Cs_4_AgBiBr_8_/WSe_2_ Heterostructure from Indirect Bandgap to Direct Bandgap by Introducing Se Vacancy"

_materials, 2023, doi:10.3390/ma16103668_

Round 1

Reviewer 1 Report

The quality of figures  should be improved.

English spell checking is required.

Author Response

I would like to express our sincere gratitude for your valuable time and effort in reviewing my manuscript. Your insightful comments and constructive feedback have helped us to improve the quality of my work and to present my research in a more effective manner.

Reviewer 2 Report

The paper "Bandgap engineering of two-dimensional double perovskite Cs4AgBiBr8/WSe2 heterostructure from indirect bandgap to direct bandgap by introducing Se vacancy" reports first-principle calculation of the electron band structures and related properties. The title of the article reflects its content. The abstract is concise and presents the main thesis of the work. The structure of the paper is concise and the scientific level is high.

There is however some correction need added to the text, e.g.

1.      Line 79: «The cut-off energy is set to 400 eV.». The choice of energy cut-off of 400 eV is not justified. Also, using the same parameters for different samples (WSe2 and Cs4AgBiBr8 monolayer) is incomprehensible.

2.      In “2. Calculation Methods” needs added information about the thickness of the studies monolayer.

3.      Figure 2. The energy band structure is given for WSe2 and Cs4AgBiBr8 monolayer. How changed the dispersion of the energy levels with the transformation from 3D (crystal) object to 2D (monolayer)?

4.      What criteria choose vacancy (Se and W) in the sample?

5.      Figure 3. Label for the figure is true? “The projected band structures (left) and the DOS (right) of the (a) Cs4AgBiBr8/WSe2 heterostructure, (b) Cs4AgBiBr8/WSe2 heterostructure with No.6 Se vacancy and (c) Cs4AgBiBr8/WSe2 heterostructure with W vacancy. The red square and blue circle in the band structures show the electron orbits of Cs4AgBiBr8 and WSe2, respectively.” If this is true, your results in fig.3 are ambiguous with fig. 5.

6.      Also, the text needs many corrections. Some of them are:

·         Article title: in formula need used sub symbols.

·         Line 6: delete “1”.

·         Line 83: “10-5 eV”, -5 must be in the power.

·         Line 84: “0.03 eV Å-1.”, -1 must be in the power.

·         Line 84: “P63/mmc” this is not a standard notation of the space group (P63mmc). The same situation to the writing of the other space groups in the text.

·         Line 84: lattice parameters need writing in italic.

In summary - I recommend this article for publication in the Materials after a major revision.

Author Response

 I would like to express our sincere gratitude for your valuable time and effort in reviewing my manuscript. Your insightful comments and constructive feedback have helped us to improve the quality of my work and to present my research in a more effective manner.Please see the attachment.

Reviewer 3 Report

The paper presents based on first-principles the study of the structure, stability, and electronic properties of 2D monolayer WSe2/Cs4AgBiBr8 heterostructure. It is found that the heterostructure does not only form type-II heterostructure with a high optical absorption coefficient, but also shows better optoelectronic properties at transformation from indirect bandgap semiconductor to direct bandgap semiconductor by introducing appropriate Se vacancy. Additionally, the stability of the heterostructure with Se atomic vacancy is discussed in different positions and found that the heterostructure is more stable when the Se vacancy is near the vertical direction of the upper Br atoms from the 2D double perovskite layer.

The significant shortcoming and missing of the paper are the following:

1. English is very poor and a whole paper must be carefully checked and revised. Only two examples:

(Line 83): “All structure were relaxed…”

(Lines 95, 101): “Lattice Parameters is…”

2. There are badly constructed or unfinished phrases:

(Lines 135, 136): “We notice that when WSe2 was adsorbed to the surface of Cs4AgBiBr8(001), there were only physical adsorption occurring and no chemical adsorption observing.”

(Lines 179 – 181): “Attempt to analysis using the GGA-PBE method on 15 different W atom vacancy heterostructures (one of which shown in Figure S1a).”

3. (Line 181): Where is Figure S1a into text?

4. (Line 225): Where is Figure S1b into text?

5. (Line 206): Where is Figure S2a into text?

6. (Lines 61, 64, 92, 160): Subscript in WSe2 must be revised.

7. (Line 84): Exponent in Å-1 must be revised.

8. (Line 66): CBM and VBM must be decrypted.

9. (Lines 108 – 113): How are obtained the transformation matrices and what are their physical mean?

10. (Line 129): “… energy of the two (monolayers?)…”

11. (Line 220): “The data are shown in Table S1”. Where is Table S1a into text?

12. All papers into References list must be accompanied by doi.

Author Response

(The authors gave the same response as above.)

Reviewer 4 Report

Attached the report

Author Response

(The authors gave the same response as above.)

Round 2

Reviewer 2 Report

The paper "Bandgap engineering of two-dimensional double perovskite Cs4AgBiBr8/WSe2 heterostructure from indirect bandgap to direct bandgap by introducing Se vacancy" reports first-principle calculation of the electron band structures and related properties of Cs4AgBiBr8/WSe2 structures. After the first revision same question about the dispersion of the energy levels are still unknown (the thickness of the layer near ~1 nm, must be given influence on dispersion of the energy levels). But, this question can be the subject of future studies. And the reasoned explanation provided (about this question) by the authors is correct.

I recommend this article for publication in Materials.

Author Response

I would like to express our sincere gratitude for your valuable time and effort in reviewing my manuscript. Your insightful comments and constructive feedback have helped us to improve the quality of my work and to present my research in a more effective manner. Meanwhile, we will continue further research on the dispersion of energy levels in our future work.

Reviewer 3 Report

Into presented new version of the manuscript, the authors have made some revision of the paper, but main issues remained to open. Therefore, I repeat them again.

1. (Line 244): Where is Figure S1a into the paper text? Any reader will not search and read Supplementary Material. Obviously, all additional figures and tables with accompanied information must be presented into the paper text. 

2. (Line 288): Where is Figure S1b into the paper text?

3. (Line 269): Where is Figure S2a into the paper text?

4. (Line 283): Where is Table S1a into text?

5. (Lines 169 – 173): The author’s Response 9 must be introduced into the paper. Moreover, the authors have replaced simply the matrices one by other. What is the right version of the matrices, namely in new or old manuscript? 
